# Emotional and Behavioural Factors Predisposing to Internet Addiction: The Smartphone Distraction among Italian High School Students

**DOI:** 10.3390/ijerph21040386

**Published:** 2024-03-22

**Authors:** Loredana Benedetto, Simone Rollo, Anna Cafeo, Gabriella Di Rosa, Rossella Pino, Antonella Gagliano, Eva Germanò, Massimo Ingrassia

**Affiliations:** 1Department of Clinical and Experimental Medicine, University of Messina, 98124 Messina, Italy; loredana.benedetto@unime.it; 2Department of Translational Biomedicine and Neuroscience, University of Bari “Aldo Moro”, 70121 Bari, Italy; 3Division of Child Neurology and Psychiatry, Department of the Adult and Developmental Age Human Pathology, University of Messina, 98124 Messina, Italy; cafeoanna@gmail.com (A.C.); rossella.pino.psy@gmail.com (R.P.); antonella.gagliano1@unime.it (A.G.); eva.germano@unime.it (E.G.); 4Division of Child Neurology and Psychiatry, Department of Biomedical and Dental Sciences and Morphofunctional Imaging, University of Messina, 98124 Messina, Italy; gabriella.dirosa@unime.it

**Keywords:** Smartphone Distraction Scale, Internet addiction, adolescents, emotional regulation, hyperactivity/inattention

## Abstract

In a digitally oriented society, smartphones provide continual online accessibility to daily life while simultaneously predisposing adolescents to engage in prolonged connections for various purposes, thus escalating the risk of Internet addiction (IA). Cognitive processes such as multitasking and attentional shifting are frequently associated with smartphone activities. Additionally, online engagements may serve as emotional strategies for regulating negative states (e.g., boredom and distress), redirecting attention towards more gratifying activities, such as social media contents. This study delves into cognitive–emotional processes (i.e., emotion regulation, attention impulsiveness, online vigilance, and multitasking) and emotional/behavioural factors (i.e., emotional problems, conduct problems, hyperactivity/inattention, peer relationships, and prosocial behaviours) that may be implicated in smartphone activities and technology addiction among adolescents. A community sample of Italian high school students (N = 676; 42.2% females) completed the Smartphone Distraction Scale (SDS), the Strength and Difficulties Questionnaire (SDQ) for internalising/externalising symptoms and the Internet Addiction Test (IAT) to assess the presence and severity of IA. The scores on the SDS were found to be positively associated with IA levels. Furthermore, students exhibiting higher internalising/externalising symptoms, particularly those with traits of attention deficit hyperactivity disorder (ADHD), are more likely to manifest problematic smartphone usage. The implications for screening adolescents more susceptible to developing IA symptoms and for implementing preventive interventions are discussed.

## 1. Introduction

Media devices have become integral to the daily activities of young individuals, with the smartphone swiftly supplanting the personal computer as the preferred means of communication (e.g., WhatsApp or Telegram messaging), social interaction (e.g., Facebook, Instagram, TikTok), learning (e.g., homework), and entertainment (such as listening to music or watching videos). It is estimated that nearly all children and adolescents (e.g., 95% in the U.S., 80% in Europe, and 90% in Asia) utilise their smartphones to access the Internet, and they report being “daily” or “almost all the time” connected online through their devices [1,2,3]. Data from an Italian survey indicate that children estimate spending over 3 h (41%) or over 2 h (29%) each day on Internet and social networking sites [4]. Nevertheless, they frequently underestimate the duration of on-screen activities, to the extent of losing track of time [5].

### 1.1. Digital Connectivity and Excessive Smartphone Usage

The increasing prevalence of smartphones, coupled with constant Internet accessibility, has resulted in a heightened incidence of Internet addiction (IA) among adolescents [6]. Montag and Reuter [7] define IA as any online-related behaviour that detrimentally affects various aspects of a user’s life, such as family conflicts, loss of friends, and diminished work/school performance. Numerous studies have consistently linked elevated daily smartphone usage to compromised family and peer relationships, challenges in school and learning, and reduced personal well-being [8,9,10]. Especially during the COVID-19 pandemic, prolonged Internet activities have been observed in young individuals, yielding adverse effects on both physical and psychological health [11,12]. However, adolescents themselves appear cognisant of the negative consequences associated with excessive engagement in online activities [13]. Results from an Italian survey indicate that distractions during homework (24%), eyes burning (21%), neck/back pain (12%), sleep disturbances (10%), and mood alterations (7%) are among the most commonly reported issues by adolescents [4]. 

Hence, the increasing body of evidence concerning the adverse outcomes associated with widespread smartphone use underscores the imperative to investigate factors linked to problematic engagement among young individuals, despite the current absence of a precise definition for “smartphone addiction”. While some authors characterise excessive smartphone use as a manifestation of Internet use disorder [14], the diagnostic criteria for Internet addiction (IA), mobile addiction, or social media addiction are currently not incorporated into the Diagnostic and Statistical Manual of Mental Disorder—5th Edition (DSM-5) [15]. Nevertheless, certain problematic behavioural patterns have been identified in the use of these devices, encompassing withdrawal (e.g., feeling anxious or irritable when the smartphone is disconnected), tolerance (i.e., escalating smartphone use), and loss of control (e.g., incessant checking or messaging), all of which detrimentally affect daily activities, emotional well-being, and social interactions. These patterns closely resemble those observed in other behavioural or substance addictions [16,17]. Furthermore, a well-established body of scientific literature attests that certain symptoms associated with IA are commonly observed in cases of excessive smartphone use, including impulsiveness, depression, anxiety, mood alterations, low self-esteem, and impaired cognitive functions [18]. Consequently, some scholars frame these symptoms in a conceptual effort to validate models that explain the onset and maintenance of behavioural addiction (e.g., [19,20]); instead, other researchers have increasingly directed their attention toward exploring the consequences of problematic engagement in online activities, rather than toward the classification or diagnostic issues (e.g., [4,21]).

### 1.2. Smartphone Distraction

Researchers agree that the advent of new technologies has fundamentally altered individuals’ interactions with their environment (e.g., [22,23]). As individuals engage in prolonged and diverse online activities, primarily through their smartphones, their attentional systems contend continually with potential overload from external factors (e.g., multitasking or focusing on notifications) and internal stimuli (such as contemplating social networks; [24]). Attention, as a cognitive process enabling individuals to process information in their environment, is inherently a limited resource. The continual use of digital devices places upon users both the demand for prolonged attentional efforts (such as watching videos) and the necessity to distribute cognitive resources among multiple stimuli simultaneously (i.e., attentional shifting). Nevertheless, protracted on-screen engagements or conflicting activities (such as checking notifications while studying or making phone calls while driving) can result in information overload, distraction, and interference with overall performance.

In accordance with Throuvala and colleagues [25], distraction is delineated as “an expression of attentional loss associated with smartphone use” (p. 12). Distraction can be characterised as a response to both external stimuli (e.g., auditory/visual cues like smartphone notifications) and internal stimuli (e.g., boredom, stress, or concerns related to news on social media), diverting the individual’s attentional resources away from ongoing activities. Distraction may also manifest when internal and external cues are conflicting, leading the individual to struggle against interferences and maintain attentional focus on pertinent stimuli (i.e., selective attention) and goal-directed behaviours. Notably, Throuvala and colleagues [25] present a multidimensional model of smartphone distraction, distinguishing its dimensions as emotive, cognitive, and behavioural processes (i.e., emotion regulation, attention impulsiveness, online vigilance, and multitasking). The underlying assumption is that digital mobile technology affords individuals various activities that support and mediate distraction processes. However, the features of smartphones (e.g., portability, connectivity, etc.) facilitate distraction, potentially contributing to problematic smartphone use, such as prolonged online activities [26].

In the model proposed by Throuvala and colleagues [25], distraction functions as a strategy for *emotion regulation* when directed towards smartphone activities that alleviate negative states, primarily boredom, anxiety, and stress. As such, distraction serves as a coping mechanism, providing relief from negative emotions, while also facilitating the experience of positive emotions, such as gratification from gaming or receiving support through social media. Consequently, maintaining a constant connection becomes a strategic approach for managing unpleasant emotions in daily life. However, this may also give rise to a potential vicious circle, sustaining excessive smartphone use [27,28]. Following this conceptual framework, negative internal states may act as precursors to both distraction and smartphone overuse [25]. In alignment with this perspective, research indicates associations between excessive smartphone use and common emotional experiences such as anger, anxiety, depression, and loneliness [18]. Particularly, during the COVID-19 pandemic, adolescents exhibited an increased disposition to use the Internet as a means of managing depressive moods and thoughts, heightening the risk of engaging in Internet abuse behaviours [29]. Additionally, Yildiz [30] found that difficulties in regulating emotions are correlated with Internet addiction (IA) and problematic smartphone use among adolescents. Similarly, in a study involving high school students, Extremera and colleagues [31] discovered that problematic smartphone users reported higher scores on maladaptive emotional regulation strategies (such as rumination, self-blame, etc.) compared to non-problematic users.

The second dimension, *attention impulsivity*, delineates a diminished control over attentive processes and an individual’s proclivity to impulsively engage in activities with the smartphone (e.g., frequent checking of notifications or scrolling through Instagram). Individuals characterised by impulsivity often display a limited ability to control their impulses, compelling them towards the immediate need to interact with the smartphone (e.g., hastily clicking “Likes”). Additionally, the mechanisms of gratification and sensitivity to immediate rewards elucidate how impulsivity predisposes and perpetuates problematic online behaviours [32]. Empirical studies affirm that impulsivity constitutes a robust individual factor correlated with behavioural addiction and excessive smartphone usage [15,26]. Furthermore, heightened attentional impulsivity has been found to be associated with overuse of social media and Facebook among adolescents [33].

*Online vigilance* is defined as a “cognitive preoccupation and orientation toward social media content” ([25], p. 13) and manifests in behaviours such as incessantly monitoring the smartphone (e.g., interrupting tasks to check messages), being on alert, or contemplating social media content. The inclination towards online vigilance is often associated with the psychological need for reassurance, particularly in maintaining social support and communication through social media channels. This mode of distraction is further heightened by the fear of missing out (FOMO), signifying the apprehension of missing out on important information or experiences within the realm of social media. Adolescents appear to be notably susceptible to FOMO, leading to a desire to uphold social connectivity, a continuous online presence, and a propensity to routinely check social media [34].

Finally, the *multitasking* dimension manifests when simultaneous activities are conducted on a smartphone (media multitasking, such as using different applications) or when diverse behaviours (e.g., walking or talking) are carried out while utilising the smartphone. Multitasking with smartphones is an exceedingly prevalent and pervasive phenomenon in daily life, to the extent that it is commonly perceived as synonymous with distraction [25], a viewpoint shared by educators and parents alike. Indeed, research demonstrates the adverse effects of media multitasking on the performance of primary tasks (e.g., students’ homework). These consequences include increased execution times (due to repeated interruptions) and elevated error rates [35]. However, adopting a more optimistic standpoint [36], not all multitasking behaviours necessarily compromise performance. The impact depends, for instance, on the type of task-switching (e.g., visual and/or verbal) or the nature of attention (i.e., focused versus distributed attention) involved. Nevertheless, given that smartphones encourage multitasking behaviours that are intermittently rewarded, individuals often find themselves engrossed in continuous digital activities, leading to heightened levels of inattention (i.e., the allocation of attentive resources across tasks) and potential performance deterioration.

In conclusion, smartphone distraction emerges as a multifaceted phenomenon [25], comprising various cognitive–emotive and behavioural components that act as precursors or predisposing factors, influencing individuals’ engagement with smartphones, while also being an outcome of their usage. Indeed, smartphone distraction is perpetuated over time due to the consequences of digital activities, such as online vigilance leading to preoccupation or constant orientation towards online notifications, interrupting attention. Simultaneously, these behaviours are reinforced by positive outcomes, such as the enjoyment of videos or engaging in friendly social exchanges. Aligned with the literature that delineates trajectories towards smartphone overuse and addiction [6,15], prolonged smartphone use has enduring implications on concentration, cognitive processes, emotional well-being, and overall performance in the long term [18,37]. Consequently, it becomes imperative to explore how the various dimensions of smartphone distraction [25], together with predisposing individual factors, contribute to the risk of developing Internet addiction (IA) among adolescent smartphone users.

### 1.3. Individual Factors and Vulnerability to IA 

It is well established that pre-existing psychological issues contribute to excessive smartphone use and, concurrently, serve as one of the pathways leading to addiction [15]. Adolescents experiencing stress, anxiety, boredom, or loneliness tend to exhibit excessive online activities and are prone to smartphone overuse [18,30,38]. Individuals engage with their mobile devices with the aim of enhancing their well-being: Internet offers immediately many amusing activities alleviating negative emotional states (e.g., anxiety or boredom), but over time, internalising problems, such as social anxiety or depression [18], may persist as a consequence of the addictive nature of the Internet. IA was also associated with impulsivity, a personality trait that can predict low inhibitory control and IA among youths and adults [39,40]. Finally, the correlation between attention deficit hyperactivity disorder (ADHD) and IA has been extensively documented in the literature (for a comprehensive review [10]). Kim and colleagues [41] found a robust link between ADHD symptoms and smartphone addiction in middle and high school students. Furthermore, the inclinations towards delay aversion, sensation-seeking, and heightened sensitivity to stimulating activities among individuals with ADHD symptoms contribute to an increased propensity for technology use, owing to the highly stimulating nature of online activities [42,43]. Therefore, attention to these individual characteristics predisposing to IA is crucial for prevention among young digital users.

## 2. Aims and Hypothesis

The main scope of this study is to identify the profile of adolescents most susceptible to developing problematic levels of Internet use through smartphones, as well as the primary predictors of their problematic involvement. To the best of our knowledge, there is a dearth of studies investigating the correlation between IA and the dimensions of distraction, as in the conceptual framework proposed by Throuvala and colleagues [25], specifically among adolescents. Accordingly, the study preliminarily seeks to confirm, within a sample of Italian adolescents, the four dimensions—namely, emotion regulation, attention impulsiveness, online vigilance, and multitasking—assessed by the Smartphone Distraction Scale [25]. Secondly, the current study seeks to explore the extent to which these dimensions are associated with IA related to smartphone usage. Finally, building on the existing literature that underscores the elevated incidence of internalising symptoms [18,41] and externalising symptoms [32,41] in problematic smartphone users, this study delves into the relationships between neuropsychological factors in adolescents and IA. To achieve the stated objectives, it is hypothesised that:(1)Elevated levels of IA will demonstrate an association with heightened levels of smartphone distraction, encompassing emotion regulation, attention impulsiveness, online vigilance, and multitasking.(2)Increased levels of emotional problems and behavioural issues (e.g., hyperactivity/inattention and conduct/interpersonal problems) will exhibit an association with elevated levels of IA.

## 3. Method

### 3.1. Participants 

The study was undertaken in the metropolitan area of Messina, located in Sicily, Southern Italy, with a community sample comprising 676 adolescents aged between 15 and 19 years (mean age = 16.94, SD = 0.92; 42.2% female). The study was a part of a screening project aimed at identifying the individual factors involved in problematic smartphone use in public high school students. The inclusion criterion mandated obtaining consent from both parents. Exclusion criteria encompassed the presence of conditions that impede the completion of self-report questionnaires, such as severe intellectual disabilities, neurological disorders, or difficulties in comprehending the Italian language by foreign students.

### 3.2. Measures

A paper-pencil battery comprising self-report questionnaires was administered to the participants.

#### 3.2.1. Descriptive on Internet Use

A set of 14 dichotomous questions formulated based on the existing literature [9,43] collected the reasons for Internet use. Examples of items include inquiries such as “When you are online, do you use Social Networking Sites?” or “When you are online, do you use streaming services?”. 

#### 3.2.2. Estimation of the Perceived Time Spent Online

Two open-ended questions were included: as in previous studies [9], participants were asked to provide information regarding the time spent online during the weekdays and weekends. Respondents indicated the perceived number of hours (in minutes) dedicated to online activities.

#### 3.2.3. Internet Addiction Test (IAT)

The Internet Addiction Test (IAT; [44,45]) was utilised to evaluate the presence and severity of Internet and technology addiction. The IAT comprises 20 items, each associated with a 5-point Likert response scale ranging from “Never” to “Always.” Sample items include inquiries such as “Did you stay online longer than you intended?” or “Do you try to hide how much time you spend online?” A total score is computed by summing the rating for each item, with the maximum possible score being 100 points. Higher scores on the IAT indicate more substantial levels of Internet addiction, with a range of 50 to 79 indicating a moderate level of addiction, and scores from 80 to 100 indicating severe addiction. In this study, the IAT demonstrated commendable internal consistency (α = 0.86).

#### 3.2.4. Smartphone Distraction Scale (SDS)

The original questionnaire [25] comprises 16 items, each associated with a 5-point Likert response scale ranging from “Very rarely” to “Very often.” The scale is designed to measure the level of distraction related to smartphone use. The items are scored to generate four factors, each consisting of 4 items: emotion regulation (ER; e.g., “Using my phone distracts me when I’m under pressure”), attention impulsiveness (AI; e.g., “I get distracted by my phone even when my full attention is required on other tasks”), online vigilance (OV; e.g., “I get anxious if I don’t check messages immediately on my phone”), and multitasking (MT; e.g., “I often talk to others while checking what’s on my phone”). To the best of our knowledge, validation in the Italian context has only been conducted on an adult sample [24]; thus, the factor structure was preliminarily verified with an adolescent sample in the present study. The SDS demonstrated an acceptable level of internal consistency for each sub-scale (α_ER_ = 0.78; α_AI_ = 0.83; α_OV_ = 0.71; and α_MT_ = 0.69). Higher scores on the scale indicate a greater level of the measured dimensions.

#### 3.2.5. Strengths and Difficulties Questionnaire (SDQ)

The questionnaire by Goodman [46,47] assesses behavioural and emotional difficulties in childhood. Comprising 25 items, the questionnaire employs a 3-point Likert scale (i.e., “Not true,” “Partially true,” and “Absolutely true”) and is structured into five factors: emotional symptoms (ES; e.g., “Complaining of headache, stomach pain, or nausea”), conduct problems (CP; e.g., “Fights with other children or annoys them on purpose”), hyperactivity–inattention (HI; e.g., “Constantly moving or uncomfortable”), peer relationship problems (PRP; e.g., “Have at least one good friend”), and prosocial behaviours (PB; e.g., “Respectful of the feelings of others”). Studies support the original factor structure of the Goodman’s questionnaire from early to late adolescence (10–19 years old, [48]), including Italian adolescents (until 18 years old, [49]). In the present study, acceptable reliability was demonstrated for each sub-scale (α_ES_ = 0.76; α_CP_ = 0.50; α_HI_ = 0.64; α_PRP_ = 0.55; and α_PB_ = 0.59). Higher scores indicate a greater level of problems for all sub-scales, except for PB dimension.

#### 3.2.6. Demographics 

Furthermore, gender and age were gathered as demographic characteristics.

### 3.3. Procedure

The questionnaires were administered in the classroom following the receipt of authorisations from the headmaster and parents. Students individually completed the questionnaires after being provided information about the research’s objectives and the assurance of response anonymity. No incentives were provided. The data were collected during the period December 2022–February 2023.

## 4. Data Analysis

The analysis was conducted in four steps.
In the entire sample, a descriptive analysis (frequencies/percentages) was conducted to identify the primary online activities engaged in by adolescents. Additionally, an estimation of the time spent online during weekdays and weekends was performed, providing mean (M) and standard deviation (SD) values as a function of gender.Then, the forms of all quantitative data distribution were examined accordingly the suggestion of [50]: if skewness and kurtosis had indexes from −1 to +1, the distributions might be considered normal. When this condition was not met, a non-parametric test would be chosen. A preliminary data analysis involved the verification of the factorial structure of the SDS scale, based on the Italian validation among adults conducted by Mascia and colleagues [24]. Confirmatory analysis was executed using diagonally weighted least squares (DWLS) estimation with the robust method of estimation, applied to compute ordinated categorical variables (i.e., Likert scales) [51]. Fit indices such as goodness-of-fit index (GFI), comparative fit index (CFI), Tucker–Lewis index (TLI), root-mean-square error of approximation (RMSEA), and standardised root-mean-square residual (SRMR) were considered for evaluating the structural model. These indexes are widely recognised in the literature pertaining to structural equation models (SEMs) [52,53,54]. Specifically, an acceptable model was considered if GFI, CFI, and TLI values approached 1, while values close to 0 were expected for RMSEA and SRMR [54]. All analyses were conducted using the Jamovi software (2.3.28.0 version) with the SEMLJ module [55]. Subsequently, IAT and SDQ scores were obtained following the relative scoring procedures proposed by Italian validations and were incorporated into the analysis as standardised measures.Pearson’s correlations were calculated to examine the associations between SDS and SDQ sub-scale scores with the IAT total score.Concerning IA, a problem group (PG; moderate or severe addiction) and a control group (CG; normal users) were identified. The selection of the two groups adhered to the cut-offs proposed by Young [44]. The PG, which constituted a sub-sample of participants reporting moderate or severe IA levels, was equated for gender and age with the CG, which was the group of normal Internet users. Subsequently, a series of analyses of variances (ANOVAs) were conducted to examine differences in the mean standard scores on SDS and SDQ. Finally, a binary logistic regression was executed to discern potential predictors of IA. For steps 3 and 4, standardised total scores were inserted in the analysis. The SPSS 26 software was employed for these stages of data analysis.

## 5. Results

Descriptively, adolescents report engaging in online activities for a variety of purposes. They are more frequently involved in the use of social networking sites, streaming, and communication services (for each of these activities, 75.1%), followed by the search for information (66%). Furthermore, adolescents converged in choosing the smartphone as the main device to being on the web (94.7%). Females report more hours (estimated in minutes) than males both in ferial days (M_Ffd_ = 428.45, SD_Ffd_ = 252.91, vs. M_Mfd_ = 345.00, SD_Mfd_ = 253.29) and during weekend (M_Fwe_ = 458.07, SD_Fwe_ = 279.24, vs. M_Mwe_ = 383.56, SD_Mwe_ = 324.15). The Mann–Whitney tests show differences between male and female on time spent online (U_fd_ = 41701.5; *p* < 0.001; U_we_ = 43761.5; *p* < 0.001).

### 5.1. Structure of Smartphone Distraction Scale

Building upon the prior validation study of the Italian SDS conducted with adults [24], the factorial structure of the SDS was examined with the adolescents enrolled in this study. The analysis affirms a structure of items organised into four factors, consistent with the original questionnaire [25] and the Italian version [24], and demonstrates acceptable fit indices: RMSEA = 0.07 [95% CI = 0.06/0.08]; SRMR = 0.06; CFI = 0.98; TLI = 0.98.

### 5.2. Correlations between IA, SDS and SDQ

Table 1 shows the Pearson’s correlation coefficients between IA and the sub-dimensions of the SDS and SDQ. Specifically, higher levels of IA significantly correlate with SDS (i.e., emotional regulation, attentional impulsivity, online vigilance, multitasking), and SDQ subscales (i.e., emotional problems, conduct problems, hyperactivity/inattention, peer relationship problems). Higher levels of IA were correlated with higher levels of SDS and SDQ subscales. Conversely, a negative correlation was found with SDQ prosocial behaviours. Higher levels of IA were correlated with lower levels of prosocial behaviours. However, the magnitude of the coefficients is low, except for SDQ hyperactivity/inattention.

### 5.3. Problematic Internet Users versus Control

#### Identification of Problematic and Control Group

Participants were differentiated based on the IAT scores as suggested by Young for the Internet use screening. Based on the proposed cut-off [44], the sample was composed of 12.6% (n = 85) normal, 64.5% (n = 437) middle, 22.5% (n = 152) moderate, and 0.3% (n = 2) severe users. To “extremise” the IA scores, participants belonging to the intermediate level (n = 437) were considered “social users”, therefore, they were excluded. Then, a subsample of 169 participants was formed: the 85 normal users were considered the control group (CG); while the problematic group (PG) was formed randomly by drawing 84 participants among those with moderate and severe levels of addiction. The CG and the PG were balanced by gender and age. CGs constituted 50.3% of the subsample (80.0% male, n = 68, and 20.0% female, n = 17) and PGs were 49.7% of the subsample (77.4% male, n = 65, and 22.6% female, n = 19). 

The ANOVAs show the comparisons between sub-groups on SDS and SDQ subscales (Table 2). In contrast to CG, the PG was characterised by higher scores on SDS emotional regulation, attention impulsiveness, online vigilance, and multitasking and SDQ emotional problems, conduct problems, and hyperactivity/inattention, whereas lower SDQ prosocial behaviours.

Table 3 presents the parameters of the binary logistic regression analysis. A significant effect of hyperactivity/inattention was identified for problematic online users (β = 0.74; *p* < 0.01). This finding indicates that higher scores on the hyperactivity/inattention dimension elevate the probability of belonging to the problematic group (PG), whereas lower scores on the same dimension characterise the control group (CG). Regarding gen-der, there is a trend toward statistical significance, suggesting that males tend to belong to the problematic group.

## 6. Discussion

The study had two main objectives: initially, to verify within an adolescents sample the structure of the SDS (Smartphone Distraction Scale) [25], a questionnaire designed to assess the cognitive-emotional processes involved in problematic smartphone use; secondly, to explore the associations between these cognitive–emotional processes, individual vulnerability factors (i.e., internalising/externalising problems), and elevated levels of Internet addiction (IA) among adolescents. As far as the authors know, no previous studies explored distraction dimensions, as operationalised in the SDS (i.e., emotion regulation, attention impulsiveness, online vigilance, and multitasking) among adolescents. Past studies revealed positive associations between these dimensions of smartphone distraction and problematic mobile use, fear of missing out (FOMO), metacognitions (e.g., advantages of smartphone use as a means to distract oneself from worries), and smartphone addiction among adult users [24,26]. Therefore, a noteworthy strength of the current study lies in its shift of focus towards adolescents, the generation characterised as both the most avid users of digital devices and the most susceptible to pathological involvement and addiction [6,15]. Consequently, recognising the cognitive–emotive and behavioural profile of adolescents potentially at risk of developing various forms of technology addiction becomes imperative for effective prevention and intervention strategies. 

The findings provide empirical support for the four-factor dimensions of the Italian SDS when applied to adolescents. Throuvala and colleagues [25] originally developed the SDS with a sample of British university students, identifying 16 items organised into 4 subdimensions (i.e., emotional regulation, attention impulsiveness, online vigilance, and multitasking). In the Italian context, Mascia and colleagues [24] conducted a validation study with adult participants, affirming a four-factor structure identical to that of the original SDS. In our study, we implemented a factorial model, which yielded results consistent with the established structure prevalent in the existing literature. Consequently, while this result allows us to apply the SDS to the adolescent sample in our study, it also underscores the need for more targeted validation research within the adolescent demographic. This dual perspective emphasises the significance of both utilising the SDS in our current investigation and promoting further validation studies tailored to the unique characteristics of adolescent populations in the Italian context.

In accordance with the hypothesis, all SDS dimensions were found to be associated with higher levels of IA, thus substantiating the assertion that adolescents with problematic engagement in online activities exhibit elevated attention interruption (i.e., multitasking), cognitive–emotive preoccupation with the smartphone (i.e., online vigilance), and employ smartphone activities as a coping strategy to alleviate negative emotional states (i.e., emotion regulation). These findings align coherently with the developmental literature regarding the psychological functions of smartphones, such as online chatting, connecting with social media networks, and engaging in activities like gambling, among others, within the context of adolescence. Specifically, adolescents consistently access social media platforms to communicate with peers, share experiences, and receive emotional support, thereby expressing a sense of connection and belongingness [56]. This motivation propels them to engage in frequent online activities. Moreover, as elucidated by Griffiths [57], the habit of using social networking sites can be comprehended by considering the impact on users’ reward systems, which tends to be unpredictable and random. An illustrative example of a reward is the receipt of “Likes” from other users, contributing to an increased “desire for validation” and prompting repeated visits to check for appreciation on social platforms. Nevertheless, intensive use of social media is correlated with adolescents’ smartphone overuse [58], contemplation of social content [59], and apprehension about losing or being excluded from enjoyable online experiences (i.e., FOMO; [60]). Our results are consistent with the studies affirming that the social use of the Internet through smartphones represents a potent source of distraction among adolescents [61].

The utilisation of smartphone distraction as a strategy for emotion regulation is also found to be significantly correlated with adolescents’ IA. In this context, smartphone distraction serves to momentarily alleviate the distress and negative mood perceived by adolescents, yet it functions as a maladaptive coping strategy that perpetuates smartphone overuse. These findings align with prior research indicating the associations between adolescents internalising problems (such as anxiety or boredom) and excessive smartphone overuse [18,30,38]. Furthermore, Marino and colleagues [59] observed that emotional regulation predicts the usage of social networking sites among adolescents. It is noted that young individuals are inclined to spend increased time on social networking sites during periods of negative emotional states. The internet is often compulsively employed with the specific aim of “altering” mood, reflecting a pattern consistent with the present study’s observations.

In the context of this study, elevated levels of attentional impulsivity—manifested as an inability to focus attention or concentrate on a primary task—were found to be associated with IA. During adolescence, a diminished attentional control can be elucidated as a correlate of incomplete brain development, resulting in observed difficulties in various cognitive tasks, including selective attention, working memory, and inhibitory control [62]. Research affirms that adolescents’ immaturity in inhibitory control, self-regulation, and attentional control is linked to smartphone addiction, with higher rates of smartphone addiction observed among children and young people aged 10–20 compared to adults aged 21–30 [63]. Particularly noteworthy, the findings from the current study contribute to existing research on the role of impulsivity [29], in conjunction with disinhibition and susceptibility to boredom (or sensation seeking [39]) as a vulnerability factor for problematic smartphone use during adolescence. 

In summary, the outcomes of the current study not only endorse the expansion of the application of Throuvala’s theoretical framework to a younger demographic of smartphone users but also endeavour to delineate a profile of individuals exhibiting problematic usage patterns. Participants of this study were community adolescents living in a hyper-connected word where the use of technological devices is pervasive, therefore a boundary between normal and problematic Internet use is very “fuzzy” [64,65]. However, due the explorative scope of this study, participants who differ with respect to levels of online engagement (lower vs. problematic) were identified according to Young’s screening criteria [44]. Following the identification of non-problematic (lower) users and those manifesting problematic levels on IAT scores, a comparative analysis of smartphone distraction dimensions and internalising/externalising symptoms was conducted across the two groups. Adolescents categorised within the problematic usage group reported more elevated levels in all dimensions of smartphone distraction, along with heightened emotional issues (e.g., anxiety, negative mood) and disruptive problems (e.g., hyperactivity/inattention and conduct/interpersonal problems). These findings contribute to the existing body of literature elucidating the association between internalising/externalising problems and problematic smartphone use during adolescence [21,41].

However, the paramount outcome of the current study lies in the identification of heightened levels of hyperactivity/inattention as a precursor to IA. These findings are in line with studies underscoring the pivotal role of impulsivity, a core symptom of ADHD, and sensitivity to rewards in the development of IA, including smartphone addiction, social media addiction, and gambling [10,66]. Adolescents exhibiting ADHD traits, particularly those with deficiencies in inhibition and cognitive control [10], may find smartphone activities exceptionally appealing, given the ability to seamlessly switch between multiple functions, respond rapidly, and receive immediate rewards. Additional studies emphasize emotional dysregulation, characterised by an inability to effectively regulate aroused emotions and the stimulation of positive affective states among individuals with ADHD [63]. These findings collectively suggest that emotional dysregulation may contribute to problematic smartphone use among individuals with ADHD.

Collectively, research corroborates the role of ADHD traits as predisposing factors for IA within non-clinical populations [40,67]. The neurodevelopmental nature of ADHD positions these traits at the extreme end of a continuum, where in a subclinical manifestation, they may also be present in the general population. Given the early onset of ADHD symptoms and the decreasing age at which individuals begin using smartphones, it is advisable to conduct further studies aimed at screening children who may be more susceptible to developing technology addiction.

### Limitations

The findings of this study are subject to certain limitations that warrant consideration. Firstly, the results are derived from a convenience sample of high school students in a city in southern Italy; therefore, generalisability across the broader Italian youth population is constrained. Secondly, due to the self-reported nature of the data, the possibility of bias in reported online activities cannot be entirely dismissed. Future investigations incorporating objective assessment methods (as exemplified by Coyne and colleagues [68]) could offer a more comprehensive exploration of the associations between smartphone distraction and various online activities (e.g., watching movies, browsing social networking sites) engaged in by adolescents. Similarly, it would be worthwhile to delve into which specific online activities (e.g., shopping, social media networking) may induce higher levels of smartphone distraction. Thirdly, internalising and externalising symptoms were self-reported by participants using the Strengths and Difficulties Questionnaire (SDQ; [46]). Although the SDQ is a widely recognised screening scale and it was employed in studies on behavioural problems associated with adolescents’ smartphone overuse (e.g., [21]), its selection aligns with the exploratory nature of the present study. However, future investigations with clinical samples could further explore the reciprocal influences between internalising factors (e.g., depression, anxiety, loneliness) or externalising disorders (e.g., ADHD) and the cognitive–emotive aspects of smartphone distraction. Specifically, the results of this study underscore the significance of high levels of hyperactivity/inattention symptoms compared to other individual characteristics (i.e., conduct problems, emotional issues, poor prosocial behaviours) in predicting Internet addiction (IA) among problematic smartphone users. Subsequent studies with clinical samples may elucidate how core symptoms of ADHD, such as impulsivity [15,69] or hyperactivity/inattention [10,67], contribute to distraction and smartphone overuse.

## 7. Conclusions

This study carries significant implications for clinical research, as well as for preventive and public health measures. A considerable proportion of adolescents exhibit a pronounced degree of Internet usage through their smartphones. Moreover, cognitive–emotive distraction processes that impede everyday functioning may come into play. As posited by Throuvala and colleagues [25], it is plausible that multiple dimensions simultaneously contribute to problematic Internet use, particularly through smartphones. For instance, a bored student (engaging in emotion regulation) may continually check their smartphone (manifesting online vigilance) while awaiting notifications, thereby interrupting the ongoing homework (engaging in multitasking). Nevertheless, individuals with problematic Internet use demonstrate elevated levels of smartphone distraction, and in the presence of heightened levels of hyperactivity/inattention, a vulnerability towards overusing the Internet via smartphones appears more probable. This holds particular relevance for clinical research objectives.

Moreover, the present study underscores the necessity of conducting screenings using comprehensive instruments (such as SDS and SDQ) to obtain increasingly precise information for subsequent interventions. While more thorough and structured validation studies with adolescents are imperative, the SDS could prove beneficial for screening and interventions within school populations [70]. It can be employed to instruct individuals on effective management of negative emotions, identification of external factors (e.g., notification cues) or internal conditions (i.e., thoughts or emotions) preceding Internet access, and the promotion of self-regulation in smartphone use. As empirically based evidence, a cognitive–behavioural intervention focusing on mindful attention, self-monitoring, and mood self-awareness demonstrated effectiveness in reducing smartphone distraction and daily smartphone usage among university students [71]. 

Hence, the present study contributes to comprehending Internet addiction as a complex, multidimensional phenomenon linked to various individual (such as impulsivity and emotional states) and contextual factors (including parenting or school learning activities) that play a role in the discomfort experienced by adolescents.

## Figures and Tables

**Table 1 ijerph-21-00386-t001:** Pearson’s *r* correlation coefficients between SDS, SDQ, and IAT measures (N = 676).

Variable	Sub-Dimension	Correlation (*r*) with IAT Total Score
SDS	Emotion Regulation	0.249 **
	Attention Impulsiveness	0.190 **
	Online Vigilance	0.204 **
	Multitasking	0.183 **
SDQ	Emotional Problems	0.327 **
	Conduct Problems	0.356 **
	Hyperactivity/Inattention	0.400 **
	Peer Relationship Problems	0.158 **
	Prosocial Behaviour	−0.130 **

Note: SDS = Smartphone Distraction Scale; SDQ = Strengths and Difficulties Questionnaire; IAT = Internet Addiction Test (total score). ** The correlation is significant at the 0.01 level (two-tailed).

**Table 2 ijerph-21-00386-t002:** Comparisons of PG versus CG (ANOVAs; means and SDs are expressed in *z* points).

Variable		Mean (SD)		*F*	*p*
		CG	PG		
SDS	Emotion regulation	−0.196 (0.574)	0.158 (0.636)	13.982	All *p*s < 0.01
	Attention impulsiveness	−0.201 (0.740)	0.160 (0.808)	8.863	
	Online vigilance	−0.153 (0.652)	0.160 (0.697)	8.772	
	Multitasking	−0.100 (0.535)	0.152 (0.584)	8.278	
SDQ	Emotional problems	−0.594 (0.923)	0.101 (0.919)	24.120	
	Conduct problems	−0.542 (0.834)	0.467 (1.239)	38.712	
	Hyperactivity/inattention	−0.696 (0.848)	0.310 (0.987)	50.534	
	Peer relationship problems	−0.222 (0.995)	0.338 (1.044)	12.730	
	Prosocial behaviour	0.271 (0.954)	−0.313 (1.060)	14.193	

Note: SDS = Smartphone Distraction Scale; SDQ: Strengths and Difficulties Questionnaire; CG = control group; PG = problematic group.

**Table 3 ijerph-21-00386-t003:** Regression coefficients (β) and relative statistics for IAT criterion variable.

Independent Variable		β	SE	Wald	df	*p* Value	Exp (β)
SDS	Emotion regulation	0.831	0.462	3.242	1	0.072	2.296
	Attention impulsiveness	0.001	0.620	0.000	1	0.998	1.001
	Online vigilance	−0.090	0.600	0.023	1	0.880	0.914
	Multitasking	−0.013	0.621	0.000	1	0.983	0.987
SDQ	Emotional problems	0.208	0.248	0.703	1	0.402	1.231
	Conduct problems	0.420	0.242	3.017	1	0.082	1.523
	Hyperactivity/inattention	0.745	0.242	9.490	1	0.002	2.107
	Peer relationship problems	0.179	0.216	0.685	1	0.408	1.196
	Prosocial behaviours	−0.187	0.221	0.720	1	0.396	0.829
Gender		−1.198	0.620	3.730	1	0.053	0.302
Constant		0.245	0.210	1.363	1	0.243	1.277

Note: SDS = Smartphone Distraction Scale; SDQ: Strengths and Difficulties Questionnaire; gender (dummy; 0 = male, 1 = female).

## Data Availability

The data presented in this study are available on request from the corresponding author.

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
