# Peer review of "Emotional and Behavioural Factors Predisposing to Internet Addiction: The Smartphone Distraction among Italian High School Students"

_ijerph, 2024, doi:10.3390/ijerph21040386_

Round 1

Reviewer 1 Report

Comments and Suggestions for Authors

The manuscript addresses an issue of great relevance to promote responsible smartphone use among the adolescent population. It aims to identify the profile of the subjects most likely to present a problematic use of the Internet through smartphones and to know the factors that are involved in this issue. However, it is necessary to review some important issues:

-Why was the metropolitan area of Messina chosen to carry out the study? What are the reasons for this choice? The authors should provide arguments justifying it.

-At what point in time were the data collected? How long did the data collection process take? These are questions for the authors to clarify.

-Three previously validated instruments were used to collect data, but 16 questions were also designed to collect information on Internet use and self-perception of Internet time. Were these questions validated? If so, in what way? No information is provided in this regard. In fact, for the definition of the first fourteen questions, it is indicated that the existing literature was consulted. Which one in particular? No bibliographic citation is provided.

-I recommend retrieving the objectives of the study at the beginning of the Discussion section.

-It is suggested that the authors of the manuscript review and adapt the citation and reference styles to the standards established by the journal.

Comments on the Quality of English Language

The quality of the English language is adequate.

Author Response

Replies to Reviewer 1

The authors thank the anonymous referee: her/his observations made it possible to improve the manuscript on several points.

Reviewer 1’s requests                                              Point-by-point Authors’ replies:

- Why was the metropolitan area of Messina chosen to carry out the study? What are the reasons for this choice? The authors should provide arguments justifying it.

This specification has been added in the text (Participants: lines 229-231):

«The study was a part of a screening project aimed at identifying the individual factors involved in problematic smartphone use in public high school students».

-At what point in time were the data collected? How long did the data collection process take? These are questions for the authors to clarify.

This specification has been added in the Procedure (lines 293-294):

«The data was collected during the period December 2022 - February 2023».

-Three previously validated instruments were used to collect data, but 16 questions were also designed to collect information on Internet use and self-perception of Internet time. Were these questions validated? If so, in what way? No information is provided in this regard. In fact, for the definition of the first fourteen questions, it is indicated that the existing literature was consulted. Which one in particular? No bibliographic citation is provided.

The bibliographic citations have been added for the 14 questions on Internet use [9, 43] and for the 2 questions on self-perceived time online [9]:

[9] Kliesener, T., Meigen, C., Kiess, W., Poulain, T. (2022). Associations between problematic smartphone use and behavioural difficulties, quality of life, and school performance among children and adolescents. BMC Psychiatry, 22(1),195-207. https://doi.org/10.1186/s12888-022-03815-4

[43] Benedetto L., Gullotta R., Ingrassia M. (2015). ADHD, sensation seeking and problematic internet use in adolescence. Psicologia Clinica dello Sviluppo, 19(3), 511-521. https://doi.org/10.1449/81779

-I recommend retrieving the objectives of the study at the beginning of the Discussion section.

-It is suggested that the authors of the manuscript review and adapt the citation and reference styles to the standards established by the journal.

Thanks for the suggestion; we reformulated the initial paragraph of the discussion to recall the objectives of the study. Please, see lines 388-393.

We did it.

Reviewer 2 Report

Comments and Suggestions for Authors

This study explores smartphone distraction but with little research implication and importance. 

The research hypothesis was not clearly justified with sufficient empirical evidence. The researchers relied heavily on one paper by Throuvala et al, and ignored many important empirical studies about smartphone distraction and problematic smartphone use in different context.

The results were poorly presented and hard to follow.

The authors were not familiar with the important theories in the topic of internet addiction and largely missed important models, such as the PIU model by Davis (2001) and the IPACE model.

The implication for this study was not clearly structured. 

Comments on the Quality of English Language

none

Author Response

Reply to Reviewer 2

Dear Reviewer,

thanks for your comments, even if they are harsh and not favorable to the study we presented.  We reply to the comments as follows:

  • «This study explores smartphone distraction but with little research implication and importance».

     Conversely, for the first time the dimensional framework of smartphone distraction, as conceptualized by Throuvala et al., was studied with adolescent people. Because the adolescents are the strongest smartphone users, also most at risk of overusing them, the results of our study are relevant for future screening and prevention programs related to the smartphone overuse (as we affirmed into Discussion).

  • «The research hypothesis was not clearly justified with sufficient empirical evidence. The researchers relied heavily on one paper by Throuvala et al. and ignored many important empirical studies about smartphone distraction and problematic smartphone use in different context».

     We disagree with the above sentence, because: a) today, the Throuvala et al.’s (2021) study is the most complete study of instrument validation related to smartphone distraction; b) the conceptual framework of Throuvala and colleagues is empirically based on robust evidence about relevant psychological constructs associated with distraction, such as in FOMO, NOMO, and so on; c) in our paper we cited six recent studies on smartphone distraction and we think that they are all relevant to. d) We agree with recognizing the presence of many models of explanation of Internet use; this data underlines the evidence that the literature is fragmented in terms of the Internet use conceptualization (Ferrante & Venuleo, 2023, https://doi.org/10.13129/2282-1619/mjcp-3016). On the one hand, the addiction models; on the other, those of the “problematic/pathological” use of the Internet. Even more, there are models explaining the Internet as a general or specific phenomenon (e.g. Davies, 2001; Caplan, 2010), but studies that investigate the relationship with intraindividual dimensions (e.g. I-PACE Model: Brand, 2019). Our study has an exploratory and preliminary character with respect to the dimensions proposed by Thouvala. The Internet Addiction measure is merely a dimension used for the level of engagement in online activities, but we are aware that adolescents live in a hyper-connected world in which they constantly are online, they use technological devices with different motives, in a world where the boundary between “normal” and “problematic” Internet use is very “fuzzy”. Specifically, Internet is part of everyday life, people carry out most of their ordinary activities online (e.g. booking tickets, shopping, email, messaging...); even more so digital native teenagers. However, PIU or smartphone addiction is not the specific research focus of our study, but the focus is on the different expressions of distraction that are involved in smartphone overuse.

  • «The results were poorly presented and hard to follow».

     Why do you say this? We don’t understand this statement: other four reviewers did not say.

  • «The authors were not familiar with the important theories in the topic of internet addiction and largely missed important models, such as the PIU model by Davis (2001) and the IPACE model».

     We repeat: our study is not strictly concerned with the theoretical model of PIU or Internet and smartphone addiction, but it is an exploratory study on smartphone distraction associated with higher IAT scores in a sample of adolescents.

     However, a note about the current debate among the scholars on the PIU or addiction models is reported in a specific paragraph (see lines 79-83). We think also that a citation of Davis, 2001 and I-PACE model is relevant to the aims of the study, and we accordingly modified the text.

  • «The implication for this study was not clearly structured».

     Why do you say this? Other four reviewers did not.

     Furthermore, we think that our Discussion and Conclusion discuss well the implications of the study, because in them we focused educational and preventive advantages linked to the study results.

Reviewer 3 Report

Comments and Suggestions for Authors

Thank you for the opportunity to review this study. In this manuscript, researchers looked into correlation between Internet Addiction and the 4 dimensions of distraction (Emotion Regulation, Attention Impulsiveness, Online Vigilance, and Multitasking). In addition, they looked into relationship between increased levels of emotional problems and behavioral issues with higher IA. Manuscript is well written and all details are documented properly. Results are interesting and definitely present valuable contribution to the current body of knowledge in this area.

I only have two minor comments:

Line 194: remove hyphen in word between

In Tables’ legends change SDS to Smartphone Distraction Scale (not Smartphone Addiction Scale)

Author Response

Reply to Reviewer 3

The authors thank the anonymous referee for appreciation, her/his observations made the revision of the text easier.

I only have two minor comments:

Line 194: «remove hyphen in word between».

                                               The error has been corrected.

«In Tables’ legends change SDS to Smartphone Distraction Scale (not Smartphone Addiction Scale)».

                                               The errors have been corrected.

                                               Thank you very much!

Reviewer 4 Report

Comments and Suggestions for Authors

It is worthwhile to predict Internet addiction in adolescents based on various cognitive-emotional processes and emotional/behavioral factors. It is believed that you designed and conducted the study well and achieved productive results. However, I found that there were some things that needed to be improved in the way of presenting the results and writing the manuscript. Here are somethings I would like you to improve it:

1. It may be possible to omit the subheadings described in the introduction as they are not specific.

2. For descriptions of measurement tools, please check other articles published in this journal and follow their format.

3. Because the participants of this study were ordinary adolescents, there were only two people in the group who were severely addicted to the Internet. So, I think it's inappropriate to divide people into groups based on Internet addiction. Therefore, it is recommended that you focus on identifying cognitive-emotional processes and emotional/behavioral variables that are correlated with Internet addiction scores and examine the predictors of IA with those variables.

4. The contents of Tables 1, 3, and 4 do not seem very meaningful. It is recommended that a manuscript submitted to journal only must only present salient findings.

Author Response

Reply to Reviewer 4

The authors thank the anonymous referee: her/his observations made it possible to improve the manuscript on several points.

Observations and replies:

  1. It may be possible to omit the subheadings described in the introduction as they are not specific.

The authors prefer to leave subheadings for two reasons: 1. they believe the subheadings help the reader, and 2. previously, in other occasions, other referees have asked to insert them into the text to organize the contents.

  1. For descriptions of measurement tools, please check other articles published in this journal and follow their format.

The manuscript was adapted at the format of IJERPH, please see rows 236-288.

  1. Because the participants of this study were ordinary adolescents, there were only two people in the group who were severely addicted to the Internet. So, I think it's inappropriate to divide people into groups based on Internet addiction. Therefore, it is recommended that you focus on identifying cognitive-emotional processes and emotional/behavioral variables that are correlated with Internet addiction scores and examine the predictors of IA with those variables.

Thank you for this comment, we think it is useful to better explain the choice of groups in the broader framework of the study. This work is a preliminary exploration of some emotional and cognitive variables with respect to levels of online engagement through smartphone use among adolescents. Based on this reason, we want to specify that the study does not have a clinical character, so this exploration does not intend to find differences between addicted adolescents (i.e., with diagnosed IA) and control group; but through the identification of groups of adolescents who differ with respect to levels of online engagement, the study aims to explore the functioning of specific individual variables.

Then, we thought it would be useful to differentiate the subjects based on the scores suggested by Kimberly Young for the Internet use screening test. For us, this grouping aims to “extremize” the responses of the participants in the IAT. Specifically, the problematic group is represented by those who positioned themselves at moderate and severe levels; the control group is made up of subjects who belonged to a lowest level of online engagement. Thus, we have excluded all subjects who placed themselves at an intermediate level, considering them as “social users”.

To explain this reason in the manuscript, we have modified the text in two points (results and discussion, respectively):

(Results, rows 364-374) “Participants were differentiated based on the IAT scores as suggested by Young for the Internet use screening. Based on proposed cut-off [44], the sample resulted in com-posed of 12.6% (n = 85) normal, 64.5% (n = 437) middle, 22.5% (n = 152) moderate, and 0.3% (n = 2) severe users. To "extremize" the IA scores, participants belonging to the intermediate level (n = 437) were considered “social users”, therefore they were excluded. Then a subsample of 169 participants was formed: the 85 normal users were considered the con-trol group (CG); while the problematic group (PG) was formed randomly by drawing 84 participants among those with moderate and severe levels of addiction. The CG and the PG was balanced by gender and age. CGs constituted 50.3% of the subsample (80.0% male, n = 68, and 20.0% female, n = 17) and PGs were 49.7% of the subsample (77.4% male, n = 65, and 22.6% female, n = 19).”.

(Discussion, rows 467-472) “Participants of this study were community adolescents living in a hyper-connected word where the use of technological devices is pervasive, therefore a boundary between normal and problematic Internet use is very “fuzzy” [Anderson et al., 2017; Caplan, 2018]. However, due the explorative scope of this study, participants who differ with respect to levels of online engagement (lower vs problematic) were identified according to Young’s screening criteria [Young, 1998].”

Anderson, E. L., Steen, E., & Stavropoulos, V. (2017). Internet use and problematic Internet use: A systematic review of longitudinal research trends in adolescence and emergent adulthood. International Journal of Adolescence and Youth, 22(4), 430-454.

Caplan, S. E. (2018). The changing face of problematic Internet use: An interpersonal approach. Peter Lang Incorporated, International Academic Publishers.

Young, K. S. (1998). Caught in the net: How to recognize the signs of internet addiction and a winning strategy for recovery. John Wiley & Sons.

  1. The contents of Tables 1, 3, and 4 do not seem very meaningful. It is recommended that a manuscript submitted to journal only must only present salient findings.

Thank you for this comment. Table 1, 3 and 4 are merely descriptive, and information could be reported in the text directly. 

Text that replaces table 1 is reported in lines 337-345.

Text that replaces tables 3-4 is reported in lines 364-373.

Reviewer 5 Report

Comments and Suggestions for Authors

Internet use among children and adolescents as well as related Internet addiction have been in a focus of researchers for the last two decades. Due to the impact of technological innovations and specific contents affecting behaviour in children and adolescents, this growing issue of Internet addiction requires further research, in order to determine the predictors.

Please, find below several comments and suggestions of mine, as follows:

Introduction

·         The major part of introduction is actually covering the work of Throuvala and colleagues as well as the description of 4 subscales of a questionnaire. You have made a good start elaborating Internet use in children and adolescents, behavioural patterns, Internet addiction, harmful effects of addiction… Both hypotheses are based on the association between IATa and constructs, therefore, the general purpose of this paper refers to the description of Internet Addiction, thus, the constructs used by Throuvala and colleagues should be indicated within that context. Contrary, in the introductory part internalizing and externalizing issues were barely mentioned only with a few sentences. I suggest you pay equal attention to each construct description.

·         Line 74-75: „These patterns closely resemble those observed in other behavioural or substance addictions.“ Reference is missing.

2. Aims and hypothesis

·         Before indicating hypotheses, consider identifying two specific goals: the first one refers to validation of structure related to the Smartphone Distraction Scale on a sample of adolescents, since that procedure has to be done before testing the hypotheses, and the second one refers to the association of IAT with the described constructs, the part where your hypothese are stated.

3.2 Measures

·         Descriptive Internet use – If I understand correctly, this questionnaire is not a scale, it is descriptive and dichotomous variables are used. In that case, I suggest that Cronbach Alpha not to be specified.

·         Strengths and Difficulties Questionnaire (SDQ) – it is indicated within scale description that it is intended for children aged 7-16 years, while elswhere in the text that it is intended for children aged 7-17 aged. Your research included adolescents up to 19 years of age. Did you make any modifications regarding the questionnaire? If yes, please state which ones? If not, refer to the author(s) who states that the application of the questionnaire without modifications is allowed on older children.

5. Results

·         There is no data on the normality of the distribution on dependent variables. In order to test the differences when using Internet by gender, you use non-parametric test (Mann- Whitney) but later on parametric tests. Please add whether you tested the normality of the distribution and what criterion was applied for analyses.

5. 2 Correlations between IA, SDS and SDQ

·         Correlations are statistically significant, but also low (except for Hyperactivity / Inattention)

5.3.1 Identification of Problematic and Control Group

·         Is there a reference for criteria definition of problematic and control group?

·         How did You balance subsampling for gender and age? Describe!

·         Line 363-365: „Regarding gender, there is a trend toward statistical significance, suggesting that males tend to belong to the problematic group.“

In the first part of the Results it was found that the schoolgirls spent more time on the Internet than schoolboys. Can we come to conclusion here that time is not an indicator of the Internet addiction?

I noticed serious reference errors:

·         The Literature list contains works  that are not cited in the paper.

·         References must be numbered in order of appearance in the text, which is not the case here.

·         Please follow:  Reference List and Citations Guide: https://www.mdpi.com/authors/references

Author Response

Reply to Reviewer 5

The authors thank the anonymous referee: her/his observations made it possible to improve the manuscript on several points.

Comments and replies:

  1. Introduction

The major part of introduction is actually covering the work of Throuvala and colleagues as well as the description of 4 subscales of a questionnaire. You have made a good start elaborating Internet use in children and adolescents, behavioural patterns, Internet addiction, harmful effects of addiction. Both hypotheses are based on the association between IAT and constructs, therefore, the general purpose of this paper refers to the description of Internet Addiction, thus, the constructs used by Throuvala and colleagues should be indicated within that context. Contrary, in the introductory part internalizing and externalizing issues were barely mentioned only with a few sentences. I suggest you pay equal attention to each construct description.

[Authors' answer]. Thanks for the suggestion, a brief review on individual factors predisposing IA has been added in the introduction (1.3 Individual factors and vulnerability to IA, please see rows 186-203). Furthermore, some references on internalizing/externalizing factors that had previously been mentioned in the discussion have been moved to this section. The following reference was added:

Choi JS, Park SM, Roh MS, Lee JY, Park CB, Hwang JY, Gwak AR, Jung HY. Dysfunctional inhibitory control and impulsivity in Internet addiction. Psychiatry Res. 2014 Feb 28;215(2):424-8. doi: 10.1016/j.psychres.2013.12.001. Epub 2013 Dec 11. PMID: 24370334.

Line 74-75: „These patterns closely resemble those observed in other behavioural or substance addictions.“  Reference is missing.

References were added:

Griffiths, M. (2005). A ‘components’ model of addiction within a biopsychosocial framework. Journal of Substance Use, 10(4), 191-197.

Kuss, D., & Griffiths, M. (2014). Internet addiction in psychotherapy. Springer.

  1. Aims and hypothesis

Before indicating hypotheses, consider identifying two specific goals: the first one refers to validation of structure related to the Smartphone Distraction Scale on a sample of adolescents, since that procedure has to be done before testing the hypotheses, and the second one refers to the association of IAT with the described constructs, the part where your hypotheses are stated.

[Authors' answer]. The authors thank the anonymous referee for this note: the text has been modified (see rows 205-213) in order to better focus on the objectives of the study.

3.2 Measures

Descriptive Internet use – If I understand correctly, this questionnaire is not a scale, it is descriptive and dichotomous variables are used. In that case, I suggest that Cronbach Alpha not to be specified.

[Authors' answer]. Thank you for the suggestion, effectively this questionnaire is not a scale, therefore the Cronbach Alpha has been removed.

Strengths and Difficulties Questionnaire (SDQ) – it is indicated within scale description that it is intended for children aged 7-16 years, while elsewhere in the text that it is intended for children aged 7-17 aged. Your research included adolescents up to 19 years of age. Did you make any modifications regarding the questionnaire? If yes, please state which ones? If not, refer to the author(s) who states that the application of the questionnaire without modifications is allowed on older children.

[Authors' answer]. A sentence (with references) justifying the use of SDQ without modification has been added:

(please see rows 281-283) “Studies support the original factor structure of the Goodman’s questionnaire from early to late adolescence (10-19 years old), including Italian adolescents (until 18 years old)”

Van Roy B, Grøholt B, Heyerdahl S, Clench-Aas J. Self-reported strengths and difficulties in a large Norwegian population 10-19 years : age and gender specific results of the extended SDQ-questionnaire. Eur Child Adolesc Psychiatry. 2006 Jun;15(4):189-98. doi: 10.1007/s00787-005-0521-4. Epub 2006 May 9. PMID: 16724172.

Corvasce, C.; Martínez-Ramón, J.P.;Méndez, I.; Ruiz-Esteban, C.;Morales-Rodríguez, F.M.; García-Manrubia,M.B. Emotional Strengths and Difficulties in Italian Adolescents: Analysis of Adaptation

through the SDQ. Sustainability 2022, 14, 6167. https://doi.org/10.3390/ su14106167

  1. Results

There is no data on the normality of the distribution on dependent variables. In order to test the differences when using Internet by gender, you use non-parametric test (Mann- Whitney) but later on parametric tests. Please add whether you tested the normality of the distribution and what criterion was applied for analyses.

[Authors' answer]. A phrase was added (lines 302-305) in order to specify the normality estimation we used.

  1. 2 Correlations between IA, SDS and SDQ

Correlations are statistically significant, but also low (except for Hyperactivity/ Inattention)

This is true: a sentence was added (line 359).

5.3.1 Identification of Problematic and Control Group

Is there a reference for criteria definition of problematic and control group?

[Authors' answer]. Yes, the groups were composed according to IAT cut-off scores (Young, 1998), and criteria are described in rows 364-374.

How did You balance subsampling for gender and age? Describe!

[Authors' answer]. The procedure for balancing the subsamples is described in rows 368-374.

Line 363-365: “Regarding gender, there is a trend toward statistical significance, suggesting that males tend to belong to the problematic group.”

In the first part of the Results it was found that the schoolgirls spent more time on the Internet than schoolboys. Can we come to conclusion here that time is not an indicator of the Internet addiction?

[Authors' answer] Yes, we can. The authors agree that the time spent online cannot constitute a discriminating criterion for Internet addiction (Musetti et al., 2016). As Musetti and colleagues (2016) point out, the understanding of the “Internet phenomenon” should be more complex and multilevel. Rollo and colleagues (2023) questioning adolescents suggest how important it is to understand the meanings underlying Internet use. Among the results, authors underline that adolescents conceptualize the problematic nature of Internet not in the use itself (quantitative assessment), but in the types of activities carried out online (qualitative view).

Musetti, A., Cattivelli, R., Giacobbi, M., Zuglian, P., Ceccarini, M., Capelli, F., ... & Castelnuovo, G. (2016). Challenges in internet addiction disorder: is a diagnosis feasible or not? Frontiers in psychology, 7, 177995.

I noticed serious reference errors:

  • The Literature list contains works that are not cited in the paper.
  • References must be numbered in order of appearance in the text, which is not the case here.

Please follow:  Reference List and Citations Guide: https://www.mdpi.com/authors/references.

We revised all  references according to Reference List and Citation Guide.

Round 2

Reviewer 2 Report

Comments and Suggestions for Authors

Thanks for the revision. 

Reviewer 5 Report

Comments and Suggestions for Authors

Thanks to the authors for considering my suggestions.

Well done! Good luck in your future scientific work!